# Do monetary incentives encourage local communities to collect and upload mosquito sound data using smartphones? A case study in the Democratic Republic of the Congo

Kieran E. Storer[1]*, Jane P. Messina[2], Eva Herreros-Moya[1], Emery Metelo[3,4], Josué Zanga[4], Nono M. Mvuama[4], Soleil Muzinga[4], Rinita Dam[5‡], Marianne Sinka[1‡], Ivan Kiskin[6‡], Josh Everett[1‡], Yunpeng Li[7‡], Stephen Roberts[8‡], Katherine J. Willis[1]

1 Department of Biology, University of Oxford, Oxford, United Kingdom, 2 School of Geography and the Environment, Oxford University Centre for the Environment, University of Oxford, Oxford, United Kingdom, 3 Department of Entomology, National Institute for Biomedical Research, Kinshasa, Democratic Republic of the Congo, 4 Kinshasa School of Public Health, University of Kinshasa, Kinshasa, Democratic Republic of the Congo, 5 Warwick Medical School, University of Warwick, Coventry, United Kingdom, 6 Surrey Institute for People-Centred AI, Centre for Vision Speech and Signal Processing, University of Surrey, Guildford, United Kingdom, 7 Centre for Oral, Clinical, and Translational Sciences, King's College London, London, United Kingdom, 8 Department of Engineering Science, University of Oxford, Oxford, United Kingdom

☉ EM, JZ, NMM, and SM contributed equally to this work
‡ RD, MS, IK, JE, YL, and SR also contributed equally to this work
* kieran.storer@biology.ox.ac.uk

## Abstract

Malaria is one of the deadliest vector borne diseases affecting sub-Saharan Africa. A suite of systems are being used to monitor and manage malaria risk and disease incidence, with an increasing focus on technological interventions that allow private citizens to remotely record and upload data. However, data collected by citizen scientists must be standardised and consistent if it is to be used for scientific analysis. Studies that aim to improve data collection quality and quantity have often included incentivisation, providing citizen scientists with monetary or other benefits for their participation in data collection. We tested whether monetary incentives enhance participation and data collection in a study trialling an acoustic mosquito sensor. Working with the community in two health areas in the Democratic Republic of Congo, we measured data collection participation, completeness, and community responses. Our results showed mixed responses to the incentive, with more participants interested in the social status and monetary value of the technology used than the monetary incentive itself. The effect of incentives on data collection varied over the course of the trial, increasing participation in the start of the trial but with no effect in the latter half of the trial. Feedback from participants showed that opinions on technology, research objectives, and incentives varied between communities, and was associated with differences in data collection quantity and quality, suggesting

**Data availability statement:** Data and analysis files are available from figshare (10.6084/m9.figshare.27332124).

**Funding:** KES, JM, EHM, EM, JZ, SM, NMM, RD, MS, IK, JE, YL, SR, and KJW were funded by the Bill and Melinda Gates Foundation Grant Award Number OPP1209888. https://www.gatesfoundation.org/ The Bill and Melinda gates foundation had no role in study design, data collection or analysis, decision to publish, or preparation of the manuscript.

**Competing interests:** The authors have declared that no competing interests exist.

that differences in community interest in data collection and the incentives may be more important than the incentive value itself. These results suggest that though there is an initial benefit, extrinsic motivations do not override differences in intrinsic motivations over time, and enhanced communication and dialogue with participants may improve citizen science participation and attitudes.

## Introduction

Mobile phone usage in sub-Saharan Africa reached 489 million users in 2023, with smartphones making up 51% of mobile phones due to improvements in mobile phone access and network connections [1–3]. With increasing smartphone ownership, their use for citizen science data collection has also increased in tandem, making use of the built-in camera, sensors, and GPS to generate and record data [4–7]. Citizen science projects have ranged widely, and have been increasingly used in healthcare settings, including for monitoring diseases such as malaria by tracking outbreaks, monitoring drug stocks, reminding patients of medical appointments, and recording mosquito larval habitats and populations [for examples see: 8–15]. As the amount and type of data collected by private citizens increases there are opportunities for developing high-resolution data repositories that widen citizen participation to include a greater range of demographic groups.

Previous research on citizen science data collection has shown that gathering data for research or healthcare purposes can help to improve educational outcomes [16], make data collectors feel more connected to their community [17], and improve health outcomes [18]. But how do researchers motivate people to participate and collect high-quality data? Firstly, there is need to understand the roles that intrinsic and extrinsic motivations play in citizen scientist participation and data collection quality. Intrinsic motivation factors are 'people's spontaneous tendencies to be curious and interested, to seek out challenges and to exercise and develop their skills and knowledge, even in the absence of operationally separable rewards' [19]. Intrinsic motivation is influenced by local and regional cultures, education, and familial values [20]. In contrast, extrinsic motivators are 'behaviours done for reasons other than their inherent satisfactions' [21], such as monetary gain [22]. Efforts to improve data collection typically target these different motivation factors. Studies have shown that monetary incentives can be used in health care and ecological monitoring schemes to improve participation [23–27] and that intrinsic factors can be harnessed to improve data collection participation and quality by improving learning opportunities and using participant knowledge [28–31].

The complex spatiotemporal variability inherent to motivation factors [32,33] and the theoretical potential for a trade-off between them [21] make understanding their influence essential to improving data collection quality and scope while optimising resource allocation. To study participants' motivations and assess the functionality of incentive applications to promote community involvement in the collection of high-quality long-term biological data, we previously ran a data collection trial using

incentives and SMS text reminders for collecting mosquito audio data in Tanzania using the HumBug tool [34]. This tool is a smartphone audio sensor combined with a modified bed net designed to temporarily trap mosquitoes and guide them towards a budget smartphone running the MozzWear app that records the flight tone of host seeking mosquitoes overnight to determine the abundance and diversity of mosquitoes present using their sound [35] (Fig 1).; For details on the Humbug tool, also see [36–42].

The smartphone app MozzWear used in the HumBug tool records mosquito flight tone data overnight and provides a secure connection to a server at the University of Oxford where the data is uploaded via the mobile or Wi-Fi data network. The data then runs through an algorithm pipeline that first detects the mosquito flight tone from background noise and then identifies the mosquito species using their acoustic signature [35–37,40]. As such, the HumBug tool is a good example of using smartphone technologies to provide an accessible biological data collection methodology to identify and monitor vectors of human disease – in this case, mosquitoes.

Previous work in Tanzania showed that providing homeowners with monetary incentives and SMS reminders to record and upload mosquito data did not significantly increase the number of uploads when compared to the control group [34]. Instead, the study found that other factors influenced data collection efforts. However, it did not examine in detail what these other factors might be. To better understand, therefore, how a monetary extrinsic factor may affect data collection

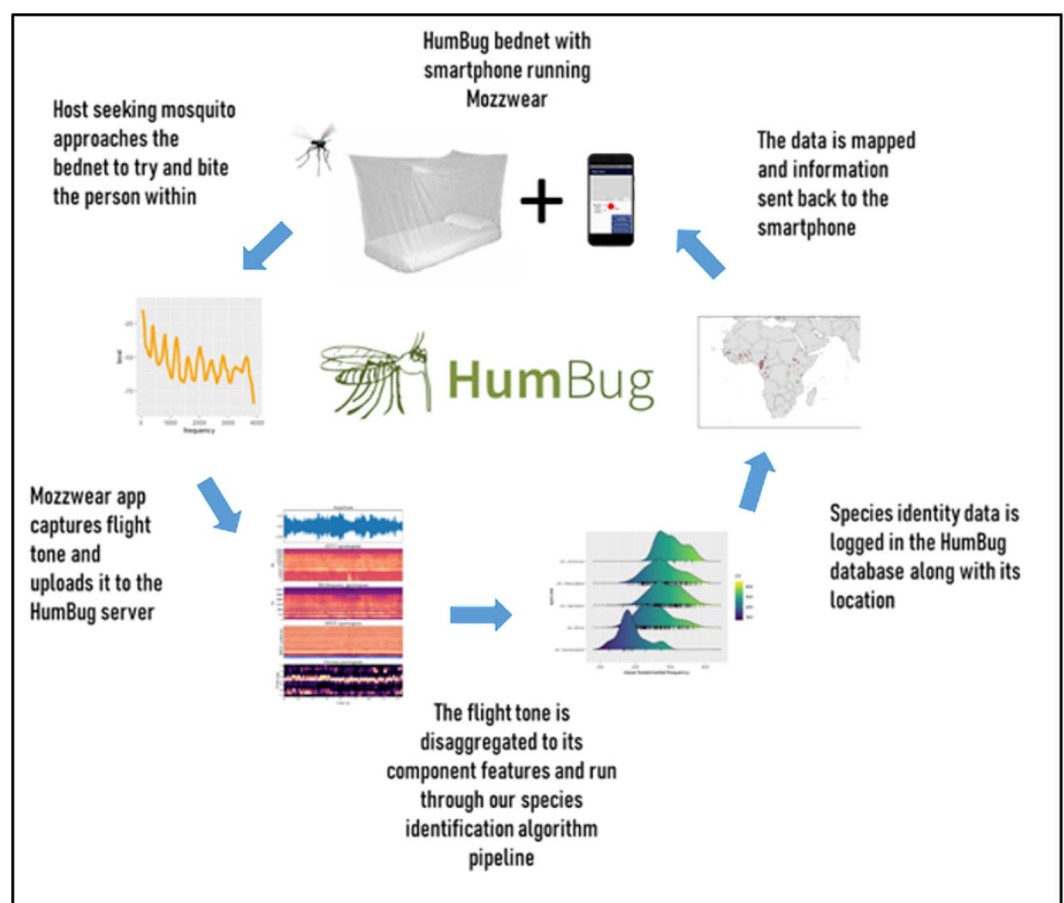

**Fig 1. HumBug Tool configuration, showing the data collection, upload, and analysis pipeline (reproduced from Sinka et al., 2021 under CC BY 4.0 license, original copyright 2021).**

practices in the use of the HumBug tool, we ran a second data collection trial within two parts of the Democratic Republic of the Congo: Bandundu, and Kinshasa. In this study we aimed to test whether monetary extrinsic incentives encouraged: i) participant data collection activity during the trial period; ii) participant effort (the number of uploads per participant made during the sampling period, indicative of following trial protocols); and iii) the persistence of participation over time (whether trial participants continued to upload data throughout the sampling period). To address these questions, we compared participant activity (weeks active) and sampling effort (number of uploads) over the sampling period to assess differences in participation associated with receiving a monetary incentive for data collection. This study builds upon our previous work in Tanzania to fill gaps in understanding how monetary incentives alone influence the consistency and quality of audio data collection.

## Methods

The study ran from April to November of 2022 in four health areas of the Democratic Republic of the Congo (DRC): two in the province of Kinshasa, and two in the province of Kwilu, near the city of Bandundu. Our collaborators at the University of Kinshasa and the University of Bandundu selected participants, conducted demographic surveys and pre- and post-trial focus group discussions, and ran the study. Data analysis and writing of the manuscript were carried out by the University of Oxford team. All leaders of communities, health zone officials, and participants provided signed consent forms for the study, and the study was approved by the University of Oxford Tropical Research Ethics Committee and the University of Kinshasa Public Health Ethics Committee. Informed written consent was obtained from all interviewees prior to the focus group discussions and the quantitative feedback survey. Consent forms signed by all the participants included the release of summary findings and details of individual responses from this study. Potential participants were informed of the voluntary nature of the study and had at least 24 hours to consider taking part. Efforts were made to create a safe place for sharing experiences during the focus group discussions.

### Study locations

Geographically, the DRC covers an area of approximately 2,345,409 km$^2$ [43]. The country is dominated by the Congo River basin surrounded by high plateaus, resulting in high precipitation and thick tropical forest within the basin and grassland in the plateaus above. In the DRC, malaria is a major cause of illness and death. Worldwide, 12% of the total malaria cases occurred in the DRC in 2022, causing 60% of hospital visits in the country [44,45]. The country contains sixteen geographic provinces which are further split into health zones (327 in total) and health areas (5357 in total) for health administrative purposes [46].

Our study took place in the DRC provinces of Kinshasa and Kwilu. The province of Kinshasa contains the capital of the DRC, the city of Kinshasa, the largest city in central Africa and the third largest mega-city on the continent, with a 2024 population estimate of approximately 17 million people [47]. The province of Kinshasa contains 420 of the country's administrative health areas [46]. The Kinshasa study sites were the Bu and Mikondo health areas of the Kinshasa province.

The Kwilu province has an estimated population of 6.6 million [48], containing 653 administrative health areas [46]. Both health areas we visited in this province, the Trois Rivières and Caravane, were part of the Bandundu health zone, which includes the city of Bandundu, the capital of the province. These study sites are hereafter referred to as the Bandundu study sites. A map showing the location of the study locations is shown in Fig 2.

Each province in our study had two trial sites: a control site and an incentive site. The selection of participants from our four trial sites took place from April 2 to June 30, 2022. In each trial site, authorities of the different health zones, the health area, and the village chiefs were contacted to gain permission for the study and to involve local leadership. Meetings with local leaders were carried out to plan the recruitment of households for the study, and to allocate which health areas would be the control treatment (Trois Rivières in Bandundu and Bu in Kinshasa) and the incentive treatment (Caravane in Bandundu and Mikondo in Kinshasa) trial sites.

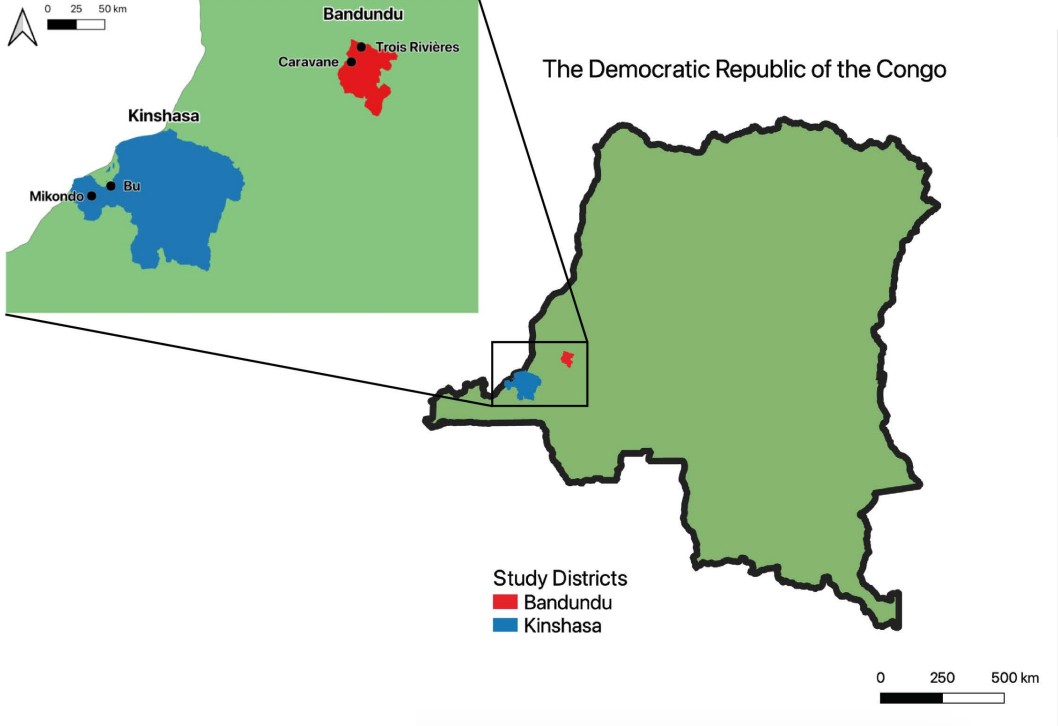

**Fig 2. Map of the Democratic Republic of Congo, modified from simplemaps.com [49] and GRID3 COD – Health Areas v4.0 [50,51] under CC BY 4.0 license, showing the location of study areas; Bandundu in red and Kinshasa in blue.** On the zoom inset, black points show the location of study sites. Mapping was carried out using QGIS 3.30.0 [52].

Households were recruited using a modified random walk technique from an entry point chosen from the main entrances to the village/street selected by local health area officials. To reduce the bias of entry point selection, each household was approached sequentially along the street and asked if they wanted to participate, until the number of households required to participate at the site (n = 37) was met. Trial participants were admissible if they were 18 + years old, owned or had access to a personal mobile phone, were residents in Bandundu/Kinshasa throughout the study, and were willing and capable of providing a signed consent form.

Ahead of the trial, training of the interviewers and moderators took place over a day in each of the trial sites to teach local health officials and local leadership how to run interviews and support participants in the trial. Demonstrators were also taught how to install and use the HumBug tool and MozzWear App and were given a presentation of the trial objectives. Finally, interviewers were trained on how to administer the demographic questionnaires to participants who agreed to take part in the trial (DHS Phase 8 Questionnaire, S1 File).

Moderators ran pre-trial focus group discussions (FGDs) to confirm that participants understood the purpose of the study, how the study was designed, and how to operate the HumBug tool and MozzWear application. The pre-trial FGDs were also used to discuss and understand the challenges that participants may face, their motivations for taking part in the study, and what they thought the impact would be on their lives. The pre- and post- trial FGD questions can be found in the S2 File. The reasons participants gave for joining the study were categorised into common themes of access to electricity, financial incentive, use of a phone, provision of bed nets, contributing to health improvements and malaria control, or gaining personal knowledge. Participant FGDs responses were then structurally coded for analysis [53].

Control group participants were provided with the HumBug tool (a smartphone running the MozzWear app and the HumBug bed net), and one dollar to pay for an internet connection. The incentive group was provided with the HumBug tool and one dollar for an internet connection, and an additional ten dollars each month paid via airtime to their mobile phones. Participants were instructed to place the smartphone with the MozzWear app in the HumBug net and start the record function at 18:00 hrs and turn it off at 06:00 hrs on their allocated weekly recording day during the trial period (16 weeks total). Recordings were split at one-hour intervals automatically in the MozzWear app to prepare recordings for algorithmic analysis. As such a complete recording effort would show 12 recordings. The recordings were then uploaded by participants to the remote server when they connected to the internet. Once participants were taught how to use the Humbug tool there was no contact between the research team and the participants.

The research team conducted post-trial FGDs to understand the participants' experience of using the HumBug tool and participating in the study. Questions included whether they liked using the MozzWear app, whether they liked using the HumBug net, how the trial personally affected them, what they would want to be different about the trial in the future, whether they would participate in the future, and if they received an incentive whether it was enough money. Data from the post-trial FGDs was coded for analysis by themes of response to questions as described for the pre-trial FGDs.

## Data analysis

To compare the effect of providing incentives to the participants to upload mosquito audio data, the number of active participants (participants who uploaded any data) and the number of uploads were counted during each trial week. Participant identification numbers were used to assess the upload counts and weekly activity by location (Kinshasa or Bandundu trial sites) and experimental group (control or incentive). All analysis was carried out in R [54]. Counts of uploads, weeks of participation, and the average number of participants each week were compared using Wilcoxon Rank Sum tests [54] and Kruskal-Wallis tests [55] with a post-hoc Dunn test [56] to assess if there was a significant difference in participation between the experimental groups/locations over the study period and reduce potential type 1 error associated with multiple comparisons. The number of participant uploads was broken down for each week of the trial to assess whether incentives improved data collection persistence throughout the study and was also compared using Wilcoxon Rank Sum Tests. Data on income, sex, age, education, and profession were also evaluated as explanatory variables to explain variation in data collection between groups. Comparisons of the demographic data of each province, trial group, and combination of the two, were made using Wilcoxon Rank Sum tests and Chi-squared tests. The pretrial FGD data was analysed using Wilcoxon Rank Sum tests to compare differences in pre-trial motivations for participation. The post-trial FGD data was analysed using a Fisher's Exact Test to assess the correlation between province and trial group on attendance for the post-trial FGD and whether participants would take part in a future trial, and phone return. All plots were produced using ggplot2 [57].

## Data management

Personal data generated in the form of signed consent forms, personal mobile phone numbers, interviews, and/or focus group audio recordings, were stored following the 2018 Data Protection Act on a secure administrative database on a University of Oxford server.

## Results

### Pre-trial focus group discussions

Pre-trial focus group discussions (FGDs) were run in three groups of 12–13 participants in each trial site (37 people per trial site, 148 participants in total). In the pre-trial FGDs, themes identified for participant motivation were monetary benefit (63.5% of all participants), contributing to science/health (33.1%), having a phone (29.7%), electricity access (27.0%) (not

a component of this study), the HumBug net (6.7%), and gaining personal knowledge (5.4%). Participants were typically motivated by multiple factors. As a participant in Bandundu describes 'To do the job well, I have to be motivated, in the sense that I have to have the material for the job first, and then I have to be paid a good payment so that I can do a good job". There was no significant difference in the number of motives each participant listed at the start of the study between Bandundu and Kinshasa (p > 0.05).

Statistical comparisons using Wilcoxon Rank Sum Tests showed that trial site participants in Kinshasa mentioned electricity significantly more as a reason to participate in the trial compared to Bandundu trial site participants (W = 3774, p-value < 0.001). Comparing treatment groups across both provinces, electricity access was mentioned more frequently in the control groups compared to the incentive groups (W = 2072, p-value < 0.001). Between the treatment groups in Bandundu, electricity was more frequently mentioned by the control group (control n = 6, incentive n = 0; W = 795.5, p-value = 0.012) and personal knowledge (on topics such as mosquito control and malaria prevention) was more commonly cited by the incentive group (control n = 0, incentive n = 4; W = 610.5, p-value = 0.042). In Kinshasa, the control group mentioned electricity (control n = 23, incentive n = 11; W = 906.5, p-value = 0.005) and contributing to science/health (control n = 15, incentive n = 6; W = 851, p-value = 0.022) significantly more than the incentive group. There were no other significant differences between provinces, treatment groups, or the treatment groups within each province (p-value>0.05).

## Demographic survey

Demographic survey data were assessed to identify if demographics contributed to differences in audio recording between provinces (Kinshasa n = 74, Bandundu n = 74), treatment groups (control n = 74, incentive n = 68), and treatment groups within provinces (Kinshasa control n = 37, incentive = 31; Bandundu control n = 37, Bandundu incentive n = 37). There were significantly more adults in the homes of participants in the Kinshasa trial sites compared to the Bandundu trial sites (W = 1680, p-value = 0.017). Between the treatment groups in both provinces, there were no significant differences in any demographic response variables. Within Kinshasa, there was significantly less water scarcity in the control group (29% experiencing water scarcity), and compared to the incentive group (45% experiencing water scarcity) (odds ratio = 0.09, p-value = 0.021) and there was a significant difference in the method of home lighting, with significantly more homes in the control group using a rechargeable flashlight, torch, or lantern (p-value = 0.016). Within Bandundu there were significantly more children in the incentive group compared to the control group (W = 675, p-value = 0.024), with one more child per house in the study on average. Additionally, there was a significant difference in toilet types between the Bandundu trial groups (p-value = 0.033). These were the only cases out of 88 comparisons that were significantly different between provinces and trial groups (S3 File).

## Comparing provinces and trial groups

Some participants decided to withdraw from the trial during its course and others failed to upload data, resulting in overall participation of 33 from the control group and 31 from the incentive group in the Kinshasa trial sites (n = 64), and 33 from the control group and 35 from the incentive group in Bandundu trial sites (n = 68), totalling 132 active participants. The withdrawal rate from the study was 7.4% from the control sites and 10.4% from incentive sites across both provinces. The reasons cited for formal withdrawal from the trial were lack of monetary benefit, unplanned move from the trial area, and loss of the smartphone. Comparisons in participation between trial provinces using Wilcoxon Continuity tests showed that Bandundu participants uploaded data more weeks of the trial period (W = 1589, p < 0.001) and had more total uploads (W = 1825.5, p = 0.009), than participants in Kinshasa, shown in Fig 3.

Comparing treatment groups across both provinces there was no difference in participation between the control group and the incentive group for either total uploads (W = 2409.5, p-value = 0.88), or participation weeks (W = 2510.5, p-value = 0.79). Within Kinshasa there was no significant difference between treatment groups in either uploads

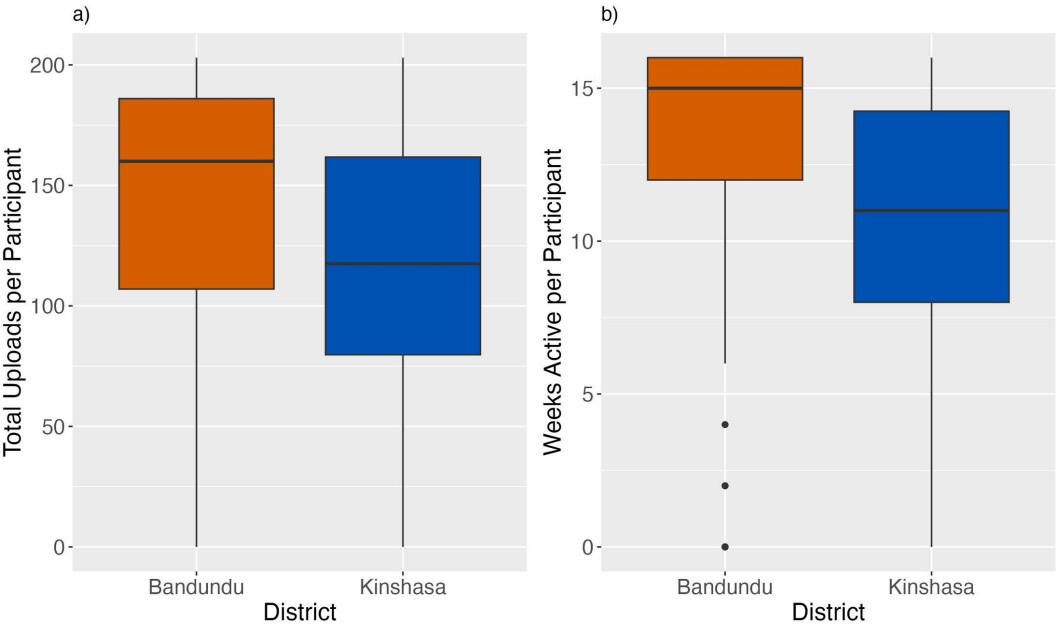

**Fig 3. Comparison of a) the total number of uploads per participant in Bandundu (n = 68) and Kinshasa (n = 64) and b) the number of weeks that each participant was active in Bandundu and Kinshasa.** Bandundu is shown in red, and Kinshasa is shown in blue. Boxes show the interquartile range (IQR), with whiskers showing the upper and lower 25% of the data, and points showing the outliers.

(W = 418.5, p-value = 0.057) or weeks of activity (W = 428, p-value = 0.073). In Bandundu, there were significantly more total uploads in the control group compared to the incentive group (W = 845.5, p-value = 0.026) but no significant difference in the average number of active weeks of each participant (W = 777, p-value = 0.13) (Table 1). There were no significant relationships found between in the demographic variables that were captured in the demographic questionnaire survey and the number of uploads or weeks of activity.

## Activity and effort over time

**Activity persistence.** The percentage of active participants was catalogued weekly to assess whether the monetary incentive affected persistence in data collection over the trial period (Fig 4). At the start of the trial, the Kinshasa incentive group had the highest participation (87.1%), followed by the Bandundu incentive (83.3%), Bandundu control (80.5%), and Kinshasa control (70.3%). Participation was approximately stable for the first half of the trial before declining from week 8. At the end of the trial both the Kinshasa control and Kinshasa incentive group had dropped to 30–35% participation, Bandundu incentive dropped to 50% participation, and Bandundu control maintained 77.7% participation.

**Data collection effort.** Data collection effort over the trial showed that in the first week the Kinshasa incentive group had the highest average uploads per active participant (12.26) followed by the Bandundu control (11.69), Bandundu incentive (10.83), and Kinshasa control (10.54). Data collection effort declined at around week seven, and at the end of the trial, the upload effort dropped for all groups, finishing in descending order with Bandundu control (11.29), Bandundu incentive (10.39), Kinshasa incentive (10.8), and Kinshasa control (9.23). Over the course of the study, Bandundu had significantly more uploads per participant compared to Kinshasa (W = 1825.5, p-value = 0.009), and the Bandundu control group had higher data collection effort than the incentive group (W = 845.5, p-value = 0.026). but there was no significant difference between the control and incentive groups overall or within Kinshasa. Fig 5 below shows the weekly comparisons in data collection effort over the course of the trial between the trial provinces, treatment types,

**Table 1. Results of the Mann Whitney-U Tests for the trial group comparisons.**

| Comparison | | Response Variable | Group | Mann-Witney U | Median | IQR | P-Value |
|---|---|---|---|---|---|---|---|
| **Province** | | **Uploads** | **Kinshasa** | W=1825.5 | 117.5 | 79.75-161.75 | 0.009486*** |
| | | | **Bandundu** | | 160 | 107-186 | |
| | | **Weeks** | **Kinshasa** | W=1589 | 11 | 8-14.25 | 0.0002879*** |
| | | | **Bandundu** | | 15 | 12-16 | |
| **Treatment** | | **Uploads** | **Control** | W = 2409.5 | 136 | 73-186 | 0.8822 |
| | | | **Incentive** | | 140 | 94.5-172.5 | |
| | | **Weeks** | **Control** | W = 2510.5 | 13 | 7-16 | 0.7852 |
| | | | **Incentive** | | 13 | 10-15 | |
| **Bandundu** | **Treatment** | **Uploads** | **Control** | W=845.5 | 180 | 122.25-193.5 | 0.02643* |
| | | | **Incentive** | | 149.5 | 91-177.75 | |
| | | **Weeks** | **Control** | W=777 | 16 | 12-16 | 0.1307 |
| | | | **Incentive** | | 13.5 | 11.5-16 | |
| **Kinshasa** | **Treatment** | **Uploads** | **Control** | W=418.5 | 99 | 62-164 | 0.05705 |
| | | | **Incentive** | | 140 | 96.5-160 | |
| | | **Weeks** | **Control** | W=428 | 11 | 6-14 | 0.07298 |
| | | | **Incentive** | | 12 | 10-14.5 | |

With * to denote significance, p-value<0.05.

*, p-value<0.01.

**. Kinshasa n=64, Bandundu n=68; Kinshasa control n=33, Kinshasa incentive n=31; Bandundu control n=33, Bandundu incentive n=35.

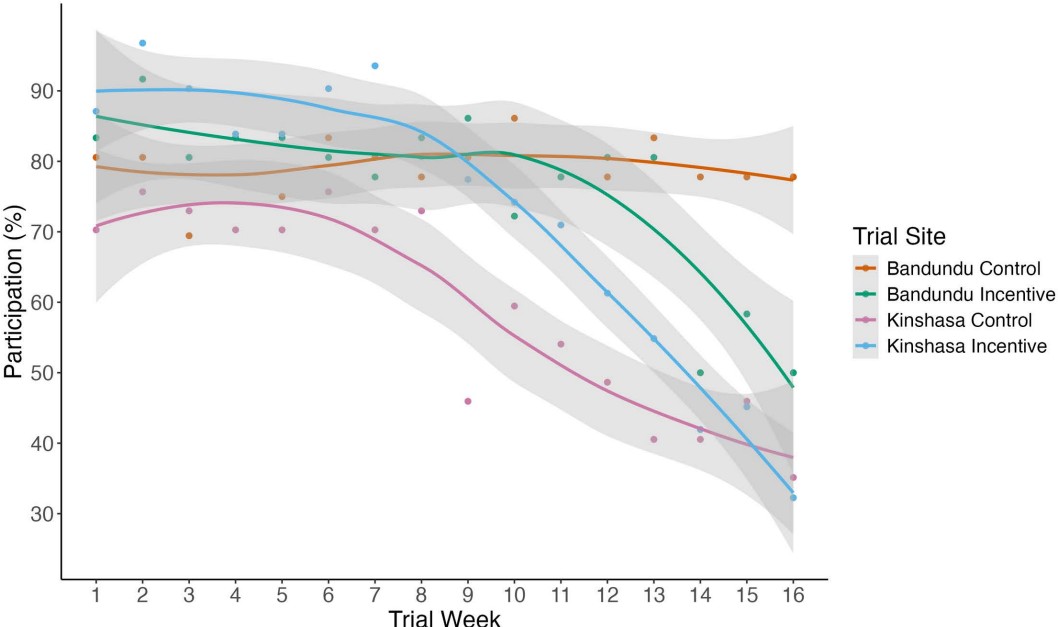

**Fig 4. The percentage of participants that activated the MozzWear app in each trial group during each week in the trial.** The coloured lines coordinate with trial groups, showing Bandundu control (n=33) in orange, Bandundu incentive (n=35) in green, Kinshasa control (n=33) in pink, and Kinshasa incentive (n=31) in blue. Lines were fitted using locally estimated scatterplot smoothing (LOESS), with the 95% confidence interval for each treatment group indicated in grey.

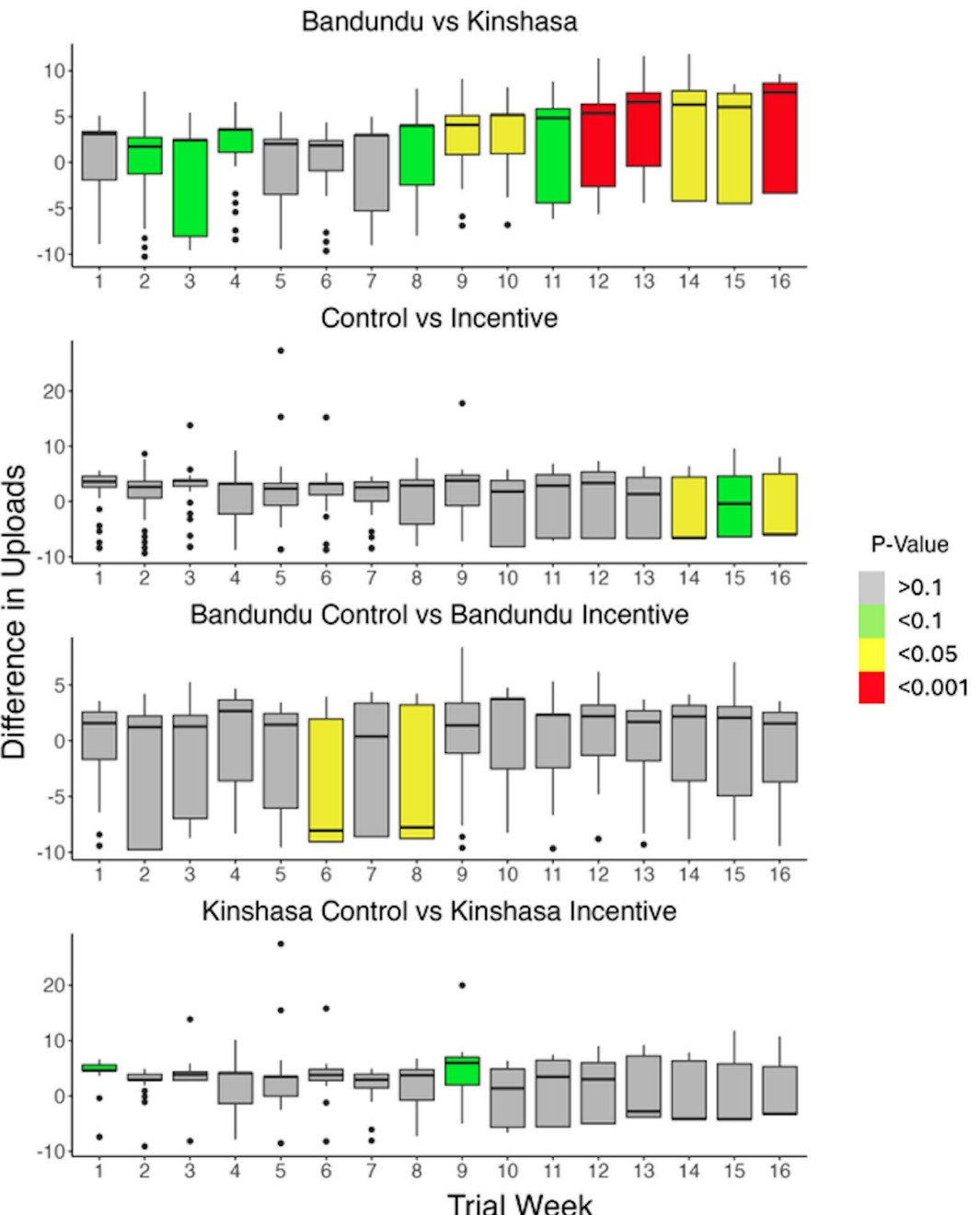

**Fig 5. Week-to-week comparisons of collection effort between the trial provinces, and incentive and control groups overall and within Kinshasa and Bandundu.** Kinshasa n = 64, Bandundu n = 68; Kinshasa control n = 33, Kinshasa incentive n = 31; Bandundu control n = 33, Bandundu incentive n = 35. The magnitude of the bars shows the difference in uploads per person from the first group in panel subtitle relative to the second. Non-significant differences are shown in grey; trends are shown in green (p-value < 0.1), significant results in yellow (p-value < 0.05), and highly significant differences are shown in red (p-value < 0.001). The whiskers around each bar show the standard error.

and treatments within provinces. Bandundu had a trend of higher uploads in weeks 2–4, 8, and 11 (p-value<0.1), and significantly higher uploads in weeks 9, 10, and 12–16 (p-value<0.05) compared to the Kinshasa groups. Comparisons of the treatment groups showed that the control groups had higher participation than the incentive groups in the final three weeks of the trial (p-value <0.05). Within Bandundu, the control group had higher participation in weeks 14 and 16 of the trial compared to the incentive group (p-value <0.05), and within Kinshasa, the control group had trends of lower participation than the incentive group in weeks 1 and 9 of the trial (p-value<0.1)

## Post-trial focus group discussions

The focus group discussions after the trial were attended by 27 participants from Kinshasa and 26 from Bandundu equating to ~35% of the original trial participants. In comparison to the 12 pre-trial FGDs, only 6 post-trial FGDs were run. Post-trial FGD attendance was lower in the Kinshasa incentive group compared to other groups (n=8, p-value=0.098, 95% CI: [0.0995, 1.2767] odds ratio: 0.368). All returning participants indicated that they had positive experiences with the HumBug sensor and a positive experience with the HumBug bed nets. When asked about the effect of the trial on their life, a respondent from the Bandundu incentive site said, "I was doing this work without someone to command me (...) it gave me the sense of responsibility."

There was a lower interest in future participation for control groups (n=12) compared to the incentive groups (n=25), and lower interest in future participation in Kinshasa (44.4% of respondents) compared to Bandundu (96.2% of respondents). Both province and trial group were significantly associated with participants interest in future participation (p-value<0.001, 95% CI: [1.801, 5.468], odds ratio: 3.102), showing that in Bandundu the control group was 10% less likely to say they would participate in the future than the Bandundu incentive group, and in Kinshasa the control group was 60% less likely to participate in the future compared to the Kinshasa incentive group.

When asked about challenges or things they wanted to change in the trial, electricity access was mentioned more frequently in the Kinshasa incentive group (W=32, p-value=0.007) and the Bandundu control group (W=129, p-value=0.008) compared to the other trial group in their respective provinces. Financial compensation/increased compensation was mentioned by 17 respondents, but did not vary significantly between trial groups. When queried on how to improve the study, a respondent indicated a desire for more interaction and information, "When we send the data, on your part, we need a sign to know if the data has arrived or not (...) until the end, we had no communication."

Finally, only participants in the Kinshasa control group mentioned wanting to keep the smartphone (n=12). Despite not being discussed in other post-trial FGDs, phone retention was an issue at the end of the study. In total, there were 33 phones retained by participants across the study, with a higher likelihood for the Kinshasa incentive group and for the Bandundu control group to retain phones, compared to the other trial group in their province (p-value=0.00516, 95% CI [1.567, 74.361], odds ratio: 9.177). Additionally, nine phones were reported as lost or stolen by the end of the trial period. As one participant phrased it from the control trial group in Kinshasa, "I want to be left with the phones because we will be laughed at (…) if you take them away from us, they will say that we participated in vain."

## Discussion

Our results demonstrate that in the DRC there was no significant difference in the number of audio recordings uploaded or the number of weeks that participants were active between the groups receiving financial incentives and the groups that did not. The previous HumBug study in Tanzania similarly showed no significant effect of incentives (monetary incentives, text message reminders, or their combination) on the total audio recordings uploaded [34]. We found there was no significant effect of demographics such as age, sex, or income on the number of uploads or participation weeks, nor were there significant differences in these demographic traits between the provinces, trial groups, or trial groups within provinces. The only significant effect on the number of uploads and active weeks was study location which showed that participation was overall higher in Bandundu compared to Kinshasa, and a difference between the control and treatment group uploads in

Bandundu due to a higher recording effort of the control group. This suggests that differences between the provinces are not attributable to demographic effects, and instead, there may be other differences, such as intrinsic motivations or community leadership, resulting in higher numbers of uploads in Bandundu compared to the Kinshasa sites. These differences may be related to the proximity of the Kinshasa health areas with the city of Kinshasa, though the access of participants to urban environments and differences in community cohesion were not specifically accounted for in our surveys.

Our study results also show that incentivisation did not significantly affect audio uploads or activity within Kinshasa and showed that the control group had more audio uploads in Bandundu compared to the incentive group. Previous studies have shown that if incentivised actions align with social norms, incentives can bolster desired behaviours [58], however, incentives can also weaken participants' intrinsic motivations—a phenomenon known as the 'crowding out' effect. This occurs when the anticipated intrinsic benefits of participation, such as the satisfaction of contributing, are diminished by the introduction of external incentives [59]. The lack of response to incentives in our study suggests that incentives were neither supporting intrinsic motivations nor diminishing them. Notably, for logistical reasons this trial used individual health areas as trial groups. Differences in response to incentives may be more strongly attributed to local culture and attitudes than treatment group. In the future, a randomised control trial that assesses participants' existing motivations could more effectively capture variations both among and within localities engaged in data collection. Additionally, further work is needed to understand what the tipping points are for incentives to significantly motivate participation, and what behaviours may either be unaffected by incentivisation or diminished by it.

Although incentives did not affect the overall number of data uploads across provinces, there were differences in the persistence of data collection participation and effort. The participation persistence of the trial groups showed that the Bandundu control group had approximately the same amount of participation from the start to the end of the trial, but that the other three trial groups experienced significant declines in the latter half of the trial. A review paper of citizen science studies found that studies that provided long term training and contact with organisers were the most successful at engaging participants [60]. The declines in robust participation towards the end of the trial in our study may be attributed to the lack of engagement with participants that was necessary to test the usability of the data collection methodology with low intervention. The low contact resulted in several post-trial focus group respondents indicating they would have liked more contact, despite all participants saying the sensor and application were easy to use. Although participants had positive attitudes regarding the data collection and responsibility (supporting results from other studies [61]) access to data and collaborators may further improve participants' feelings about their participation and their contributions [62].

Participant-researcher contact may be why our previous trial in Tanzania found that text reminders, both alone and paired with an incentive, positively affected persistence in data collection, despite the monetary incentive alone not showing a significant effect in that study system [34]. The declines in participation persistence and effort seen in this study support this previous finding that incentives alone are not enough to motivate participation. The tipping point in the middle of the study where participation dropped suggests that participants do not need further encouragement or motivation in the short term to collect Humbug data. In longer term contexts however, consistent contact and communication may be necessary to continue to motivate participants, and we have found that monetary incentives do not accomplish this, and in fact appear to only increase participation in the early phases of the study.

The differences in upload effort between our trial groups did not follow patterns relating to incentivisation. The data collection effort in Bandundu shows that the control group performed similarly to the incentive group for most of the trial, with the control group performing significantly better in the final couple of weeks. This appears to be due to a continued high level of data collection persistence and effort on the part of the Bandundu control group, which seems to be attributable to some intrinsic difference in this site compared to the other sites. In Kinshasa, incentivisation improved data collection efforts in the early parts of the trial compared to the control group, suggesting that at the time when participants were overall the most active, the incentive did improve the completeness of data collection efforts. However, this higher level of data collection effort by Kinshasa's incentive group dropped approximately halfway through the trial, after which there was

not a significant difference between the control and incentive groups. This mirrors the results for participant activity, which showed that participation significantly dropped off halfway through. The decline in data collection efforts shows that even those who are continuing to participate are not collecting data for the entire time period (6 pm to 6 am) as requested in the data collection methodology. This may be in part due to the design of our incentives, which were given regardless of activity and were not scaled for completeness of participation. This result shows that despite incentivisation initially appearing to motivate complete data collection, the diminishing interest and effort over time are not overcome by incentivisation.

Our post-trial FGDs gave insight into issues surrounding incentives, including compensation and technology. Over half of the participants in the trial did not return for the post-trial FGDs. The reasons cited by the DRC team were a desire to keep the smartphone and the lack of payment at the control sites, though there were more post-trial FGD attendees from the control groups than the incentive groups. This compensation issue was emphasised by a refusal to return the phones by some participants in trial groups in both provinces. The Bandundu control group exhibited higher phone retention compared to the Bandundu incentive group, and the Kinshasa incentive group showed higher phone retention in contrast to the Kinshasa control group. There was no significant difference in the number of participants that named phone retention as a motive for participation between the provinces, the trial groups, or between the trial groups within each province, showing that previously held values were not associated with phone retention. Instead, the pattern of phone retention mirrored the activity levels of each trial group, perhaps indicating that the more active participants felt a stronger entitlement to keeping the phones. However, phone retention was related to the relative effort in each province, rather than the overall effort. For example, the Bandundu incentive and the Kinshasa incentive had similar activity levels and total uploads, yet the Kinshasa incentive group retained twice the number of phones. As far as we are aware, participants in each province and trial group did not know the participation rates of the other groups, and therefore, did not know their relative participation. This indicates that individuals' participation might be influenced by their perception of effort.

The phone retention and refusal to return at the end of the study is an example of how citizen science projects can deviate from planned outcomes. In this context, participants effectively provided themselves with an incentive (or supplementary incentive) by refusing to return phones, despite study agreements and information about compensation ahead of the trial. A study on citizen science with specially designed video games found that game design can diminish citizen science outcomes and enhance the odds of cheating and cutting corners, depending on the ability to do so [63]. It is important to consider how incentive application design therefore may influence study outcomes and participation. The incentive group in this case received payment regardless of participation, which may have opened this study system to cutting corners (e.g., reduced activity and upload effort at the end of the study period). As an alternative, incentives given in response to data collection activity close this loophole and effectively encourage data collection persistence. Additionally, the post-trial survey in this study was not incentivised, and therefore participants gained more from refusing to return phones than they did from attending. Balancing incentive value with requests and alternatives is necessary for incentives to operate as intended, and as seen in this study, some individuals will not follow prescriptive actions if they feel they can benefit in other ways.

Our post-trial FGDs revealed that despite there not being a significant or differential response to incentivisation, there were significant contrasting attitudes between the study groups in Bandundu and those in Kinshasa. Bandundu exhibited a significantly higher overall participation rate in FGDs compared to Kinshasa, and a greater proportion of Bandundu participants expressed willingness to partake in the trial again, irrespective of incentivisation (25 out of 26 post-trial FGD attendees). Out of 27 post-trial FGD attendees from Kinshasa, only 12 indicated a willingness to participate again. Additionally, there appear to be differences in values between the two provinces. Among the 16 participants who identified knowledge as their motive for future participation, only one was from Kinshasa. Conversely, among Kinshasa participants who expressed willingness to participate again, the majority cited incentives as their driving factor. Further research is needed to understand how community and culture interact with incentive application, but in this case, despite differing motivations for participation, incentives did not have any significant effects on overall participation in the study.

The biggest issues cited by participants who took part in the trial were access to electricity to charge the smartphone sensor and internet connection. Interestingly, the groups that mentioned access to electricity significantly more were also the groups that performed best for data collection activity and effort in each province. It could be that the practicality of it was more apparent to the groups that put more effort in, and practical difficulties may be part of the reason people felt they should have been compensated better for the study. Of the incentive group participants who attended the focus group, over half said they felt they had not been given enough money for the study (14/22). This dissatisfaction may have been due to the unexpected cost of transferring money, which reduced the originally stated compensation (from $10 to $9). In pre-trial FGDs, there were only six cases when participants who cited money as their motivation asked for a monetary value at or below the incentive value given in the study, suggesting that the incentive potentially fell below common expectations.

Despite the sentiment that the money was not enough, incentive group participants were more likely to say they would participate in the future than control group participants, and although bed nets were one of the least cited reasons for participating in the study, almost all participants mentioned the bed nets as a benefit they experienced in the trial, perhaps improving their desire to continue participation. Considering that all phones were turned on and collected data at least once during the study period, it appears useability was not a limiting factor of the study, and indeed all participants apart from one said that they felt the MozzWear app was easy to use. Due to the lack of practical issues with the study, our results highlight the importance of scaling expectations of citizen science and matching the value of incentives with the effort and length of trials to improve participation and satisfaction.

Our findings show that mobile applications using hardware on budget smartphones can generate useful data with limited interventions and suggests that mobile application development should be more widely considered in citizen science data collection efforts. Future research on the role of incentivisation for data collection that utilises technology should consider how to further disambiguate the inherent benefits in participation related to technological access, such as social status or monetary benefit from retaining equipment. We suggest that with increased mobile phone ownership, this complexity may be overcome by utilising current phone owners, rather than providing participants with hardware. Additionally, as mobile phone use becomes more popular, opportunities for large scale studies will support more complex statistical approaches to account for the complex structures that underpin intrinsic motivators and responses to extrinsic motivators, such as proximity to urban areas which are potentially obscured by the inferential statistical methods used in this study.

## Conclusions

Results of this study show that in this study system incentives did not have significant effects on data collection activity throughout the trial but did have some effect on data collection persistence and effort, improving data collection effort during portions of the trial in one of our study sites. Our results indicate that community differences are what drive the trends in participation in this study system, and that more information is needed ahead of trials to assess how community differences may impact participation in citizen science data collection. Participant engagement with researchers and study organisers appears to improve data collection, particularly in longer studies. Communication with participants is important for appropriately scaling incentives to account for citizen data collection effort. Systems should be designed appropriately to maximise the alignment of incentives with active data collection. Overall, the incentives tested do not appear to significantly improve data collection in this study system, and in this case, increased engagement during data collection may have improved outcomes. With technology becoming an increasingly used tool in citizen science, its application should be considered in reference to existing infrastructure. Consideration should be made for the value that the technology represents in reference to the overall value of the incentive being offered to avoid issues with data sensor retention in response to changing attitudes towards incentive value.

## Supporting information

**S1 File. The DHS Demographic Survey Administered to Participants.**
(PDF)

**S2 File. The focus group discussion questions administered pre-trial and post-trial.**
(PDF)

**S3 File. A table of results from comparisons of the demographic survey between study provinces, trial groups overall, and trial groups within each province.**
(PDF)

## Acknowledgments

Thank you to the DRC team at Kinshasa School of Public Health and University of Bandundu and to Paul Mansiangi, who has since passed away, for organising and co-ordinating the study. We are grateful to the community leaders, health zone officials, and all the participants from Trois Rivières, Caravane, Bu and Mikondo for their support and participation in the study.

## Author contributions

**Conceptualization:** Eva Herreros-Moya, Emery Metelo, Josué Zanga, Nono M. Muvamba, Soleil Muzinga, Rinita Dam, Marianne Sinka, Ivan Kiskin, Josh Everett, Yunpeng Li, Stephen Roberts.

**Data curation:** Kieran Storer.

**Formal analysis:** Kieran Storer, Jane P. Messina.

**Funding acquisition:** Katherine J. Willis.

**Investigation:** Eva Herreros-Moya, Emery Metelo, Josué Zanga, Nono M. Muvamba, Soleil Muzinga.

**Methodology:** Kieran Storer, Eva Herreros-Moya, Emery Metelo, Josué Zanga, Nono M. Muvamba, Soleil Muzinga, Rinita Dam, Marianne Sinka, Ivan Kiskin, Josh Everett, Yunpeng Li, Stephen Roberts, Katherine J. Willis.

**Project administration:** Eva Herreros-Moya.

**Supervision:** Eva Herreros-Moya, Katherine J. Willis.

**Visualization:** Kieran Storer, Jane P. Messina, Eva Herreros-Moya.

**Writing – original draft:** Kieran Storer.

**Writing – review & editing:** Eva Herreros-Moya, Emery Metelo, Josué Zanga, Nono M. Muvamba, Soleil Muzinga, Rinita Dam, Marianne Sinka, Ivan Kiskin, Josh Everett, Yunpeng Li, Stephen Roberts, Katherine J. Willis.

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
