## [Decision Letter · Decision Letter 0]

11 Mar 2025

PONE-D-24-49851What incentives encourage local communities to collect and upload mosquito sound data by using smartphones? A case study in the Democratic Republic of the CongoPLOS ONE?

Dear Dr. Storer,

Thank you for submitting your manuscript to PLOS ONE. After careful consideration, we feel that it has merit but does not fully meet PLOS ONE’s publication criteria as it currently stands. Therefore, we invite you to submit a revised version of the manuscript that addresses the points raised during the review process.

We look forward to receiving your revised manuscript.

Kind regards,

Raquel Inocencio da Luz, Phd

Academic Editor

PLOS ONE

Journal Requirements:

4. We note that Figure 2 in your submission contain map/satellite images which may be copyrighted. All PLOS content is published under the Creative Commons Attribution License (CC BY 4.0), which means that the manuscript, images, and Supporting Information files will be freely available online, and any third party is permitted to access, download, copy, distribute, and use these materials in any way, even commercially, with proper attribution. For these reasons, we cannot publish previously copyrighted maps or satellite images created using proprietary data, such as Google software (Google Maps, Street View, and Earth). For more information, see our copyright guidelines: http://journals.plos.org/plosone/s/licenses-and-copyright.

5. We note that Figure 3 and 4 includes an image of a participant in the study.

Reviewers' comments:

Reviewer's Responses to Questions

**Comments to the Author**

1. Is the manuscript technically sound, and do the data support the conclusions?

Reviewer #1: Yes

Reviewer #2: Yes

2. Has the statistical analysis been performed appropriately and rigorously?

Reviewer #1: I Don't Know

Reviewer #2: Yes

3. Have the authors made all data underlying the findings in their manuscript fully available?

Reviewer #1: Yes

Reviewer #2: No

4. Is the manuscript presented in an intelligible fashion and written in standard English?

Reviewer #1: Yes

Reviewer #2: Yes

Reviewer #1: This manuscript, titled ''What incentives encourage local communities to collect and upload mosquito sound data by using smartphones? A case study in the Democratic Republic of the Congo'', investigates the effect of monetary incentives on citizen science participation in the Democratic Republic of the Congo (DRC), using acoustic mosquito sensors as a tool for data collection. The study builds on previous work conducted in Tanzania and explores both urban and rural settings, and provides insights into the role of intrinsic motivations versus extrinsic incentives in sustaining community engagement.

The manuscript is strong and well-written, with appropriate ethical considerations and study design. However, there are areas where the manuscript could be improved to enhance clarity, ease of interpretation, and broader impact. Specific feedback is provided below.

Major comments:

Lines 79-85: the introduction references the Tanzania study but does not clearly position this manuscript as a follow-up. It could strengthen the manuscript to explicitly state how this work builds on or how it differs from the Tanzania findings. It is recommended that the authors add a sentence explicitly linking the gaps identified in the Tanzania study (for instance, rural-only context) to the objectives of this work.

Regarding the Focus Group Insights, lines 247-253, the discussion of Focus Group Discussions (FGD) results is useful but could benefit from additional depth. For example, higher persistence in Bandundu could be contextualised using participant quotes. A suggestion would be that the authors add representative quotes from FGDs to illustrate key themes and link them to the observed participation trends.

The study's inclusion of Bandundu (rural) and Kinshasa (urban) appear to be an opportunity to explore how rural-urban differences influence participation. However, the manuscript does not fully address these dynamics or their potential implications. In lines 401-402, the authors note that differences in participation are not attributable to demographic factors but do not explore other potential influences, such as rural-urban dynamics that may be marked in the two setting of the study. This section could be expanded on to discuss e.g. infrastructure, community cohesion, and leadership differences between Bandundu and Kinshasa.

Minor comments:

Line 46: replace "effecting sub-Saharan Africa" with "affecting sub-Saharan Africa."

Discussion: the authors acknowledge that the findings are context-specific but do not discuss how they might be generalised to other settings. The paper could benefit from expanding the discussion to include how differences in infrastructure, culture, or community norms might affect the scalability of the study’s approach. Ethical considerations, such as phone retention, may also be under-discussed in terms of their potential impact on the findings or on the design of the study.

Future Research Directions: the conclusion could benefit from a brief discussion of future research priorities, such as testing alternative incentive models, evaluating long-term engagement strategies, and/or assessing the usability of the acoustic mosquito sensors in the study context.

Methodology improvement: the supplementary file provides detailed documentation of participant demographics and focus group questions, which supports the reproducibility of the study. Nevertheless, the modified random walk technique used for participant selection is not sufficiently explained in either the manuscript or the supplementary materials. While the inclusion of demographic comparisons and FGD protocols is quite valuable, the lack of detail about the recruitment method limits the transparency of the methodology. For instance, how were starting points for recruitment selected? Were any randomisation procedures applied to maintain the sampling unbiased? Additionally, minor gaps in the description of the design of text message reminders could also limit reproducibility. It is recommended that the authors provide additional detail or supplementary materials for these two aspects.

Data availability seems to be in compliance with PLOS guidelines. Still, it is important to confirm that the repository cited by the authors includes raw data points (such as the individual participation records and anonymised FGD transcripts), as I was not able to access it. The statistical methods used appear to be appropriate but I would recommend consulting someone with higher expertise in this matter. The methodology is sufficiently detailed overall. However, minor gaps in the description of participant recruitment (e.g., modified random walk technique) and the design of text message reminders could be further improved.

In conclusion, the manuscript is suitable for publication following few major and minor revisions. Addressing the comments above will improve the manuscript’s clarity and overall contribution to the fields of malaria research and citizen science.

Reviewer #2: This study has addressed an important question about the influence of monetary incentives on participation (data collection) in citizen research. The methodology and results have been excellently presented and discussed. The researchers have done an excellent job of using quantitative and qualitative methods to address their study questions. The manuscript is indeed of very high quality.

Here are a couple of comments that I recommend that the authors consider:

1) The topic of the study is not congruent with the study objective or question. The title “What incentives encourage local communities to collect and upload mosquito sound data by using smartphones? A case study in the Democratic Republic of the Congo” suggests that the study has evaluated different type of incentive while it has specifically assessed the influence of “monetary incentives” A title that would fit better with the study objective/question/scope could be “Do monetary incentives encourage local communities to collect and upload mosquito sound data by using smartphones? A case study in the Democratic Republic of the Congo”

2) The researchers may have made efforts to adjust for and discuss confounding inherent on the research design but need to describe this more clearly in the report. Also, I recommend that authors highlight known limitations of the main inferential statistical approach used in comparative analysis.

**Do you want your identity to be public for this peer review?** For information about this choice, including consent withdrawal, please see our Privacy Policy

Reviewer #1: No

Reviewer #2: **Yes: ** Martin Meremikwu

---

## [Author Response · Author response to Decision Letter 1]

28 Apr 2025

Thank you for the opportunity to submit a revised draft of our manuscript titled “What incentives encourage local communities to collect and upload mosquito sound data using smartphones? A case study in the Democratic Republic of the Congo” to PLOS One. We appreciate the time you and the reviewers have taken to provide valuable feedback and insightful comments. We have made changes to incorporate most of the suggestions provided by reviewers and they are highlighted through the manuscript. Please find below the point-by-point responses and edits made to the paper in line with feedback.

Journal Requirements:

Comment 1: Please ensure that your manuscript meets PLOS ONE's style requirements, including those for file naming.

Response: The manuscript has been amended to address the style and naming requirements.

Comment 2: Please note that funding information should not appear in any section or other areas of your manuscript. We will only publish funding information present in the Funding Statement section of the online submission form. Please remove any funding-related text from the manuscript.

Response: The manuscript has been amended to remove funding information from the manuscript.

Comment 3: When completing the data availability statement of the submission form, you indicated that you will make your data available on acceptance. We strongly recommend all authors decide on a data sharing plan before acceptance, as the process can be lengthy and hold up publication timelines. Please note that, though access restrictions are acceptable now, your entire data will need to be made freely accessible if your manuscript is accepted for publication. This policy applies to all data except where public deposition would breach compliance with the protocol approved by your research ethics board. If you are unable to adhere to our open data policy, please kindly revise your statement to explain your reasoning and we will seek the editor's input on an exemption. Please be assured that, once you have provided your new statement, the assessment of your exemption will not hold up the peer review process.

Response: All the raw data from our study is now publicly available at 10.6084/m9.figshare.27332124

Comment 4: We note that Figure 2 in your submission contain map/satellite images which may be copyrighted. All PLOS content is published under the Creative Commons Attribution License (CC BY 4.0), which means that the manuscript, images, and Supporting Information files will be freely available online, and any third party is permitted to access, download, copy, distribute, and use these materials in any way, even commercially, with proper attribution. For these reasons, we cannot publish previously copyrighted maps or satellite images created using proprietary data, such as Google software (Google Maps, Street View, and Earth). For more information, see our copyright guidelines: http://journals.plos.org/plosone/s/licenses-and-copyright.

Response: To comply with the rules regarding CC BY 4.0 the original data showing the health districts of the DRC from the Humanitarian Data Exchange (DRC health Data) was removed and replaced using Creative Commons Attribution Licensed data. The Democratic Republic of the Congo (DRC) GIS data from simplemaps, the GRID3 COD - Health Areas v4.0: Kwilu health areas (GeoPackage), and the GRID3 COD - Health Areas v4.0: Kinshasa health area (GeoPackage) are all accessible under existing CC BY 4.0 licenses. This change in data used in the map of the DRC has also been reflected in the addition of two references for the GeoPackages used, along with the statement of origin in the caption on Figure 2:

Center for International Earth Science Information Network (CIESIN), Columbia University, Ministère de la Santé Publique, Hygiène et Prévention, Democratic Republic of the Congo, and GRID3. (2025). GRID3 COD - Health Areas v4.0: Kinshasa health areas (GeoPackage). New York: Columbia University. https://data.humdata.org/dataset/grid3-cod-health-areas-v4-0. Accessed April 3, 2025.

Center for International Earth Science Information Network (CIESIN), Columbia University, Ministère de la Santé Publique, Hygiène et Prévention, Democratic Republic of the Congo, and GRID3. (2025). GRID3 COD - Health Areas v4.0: Kwilu health areas (GeoPackage). New York: Columbia University. https://data.humdata.org/dataset/grid3-cod-health-areas-v4-0. Accessed April 3, 2025.

Comment 5: We note that Figure 3 and 4 includes an image of a participant in the study.

Response: These images have been removed from the manuscript.

Comment 6. Please review your reference list to ensure that it is complete and correct. If you have cited papers that have been retracted, please include the rationale for doing so in the manuscript text or remove these references and replace them with relevant current references. Any changes to the reference list should be mentioned in the rebuttal letter that accompanies your revised manuscript. If you need to cite a retracted article, indicate the article’s retracted status in the References list and also include a citation and full reference for the retraction notice.

Response: The citation list has been reviewed and there were no instances of retracted papers. The reference list was amended with the GeoPackage references for the Kwilu and Kinshasa health districts (shown above), and the references were renumbered to reflect these additions.

Comments from Reviewer 1

Major comments

Comment 1: Lines 79-85: the introduction references the Tanzania study but does not clearly position this manuscript as a follow-up. It could strengthen the manuscript to explicitly state how this work builds on or how it differs from the Tanzania findings. It is recommended that the authors add a sentence explicitly linking the gaps identified in the Tanzania study (for instance, rural-only context) to the objectives of this work.

Response: This study is named directly as a second trial of the mosquito detection application with monetary incentives in line 187. Verbiage specifying the research’s focus on the use of monetary extrinsic factors, and how the study builds on the study in Tanzania has been added in lines 205-206, stating: ‘This study builds upon our previous work in Tanzania to fill gaps in understanding how monetary incentives alone influence the consistency and quality of audio data collection.’

Comment 2: Regarding the Focus Group Insights, lines 247-253, the discussion of Focus Group Discussions (FGD) results is useful but could benefit from additional depth. For example, higher persistence in Bandundu could be contextualized using participant quotes. A suggestion would be that the authors add representative quotes from FGDs to illustrate key themes and link them to the observed participation trends.

Response: Due to the lack of significant differences in pre-trial FGD responses between Kinshasa and Bandundu it does not feel appropriate to use a FGD response in this section of the results to highlight differences in the groups that were seen later in the trial (such as in the post-trial FGDs). Instead, a quote from a participant listing multiple motivation factors has been added in lines 373-377 to demonstrate how complex participant motivations were prior to trial participation.

Comment 3: The study's inclusion of Bandundu (rural) and Kinshasa (urban) appear to be an opportunity to explore how rural-urban differences influence participation. However, the manuscript does not fully address these dynamics or their potential implications. In lines 401-402, the authors note that differences in participation are not attributable to demographic factors but do not explore other potential influences, such as rural-urban dynamics that may be marked in the two setting of the study. This section could be expanded on to discuss e.g. infrastructure, community cohesion, and leadership differences between Bandundu and Kinshasa.

Response: This study was not aimed at specifically addressing urban versus rural factors for responses to incentivization, and the Kinshasa study sites were not located in the city but in rural parts of the health district (referred to as rural Kinshasa when first introduced in the manuscript, line 188). However, we acknowledge the potential role of proximity to urban areas and have now addressed this in the manuscript discussion lines 547-549, reading: ‘These differences may be related to the proximity of the rural Kinshasa sites with the city of Kinshasa, though the access of participants to urban environments and differences in community cohesion were not specifically accounted for in our surveys.’

Minor comments:

Comment 4: Line 46: replace "effecting sub-Saharan Africa" with "affecting sub-Saharan Africa."

Response: This has now been amended in the manuscript.

Comment 5: Discussion: the authors acknowledge that the findings are context-specific but do not discuss how they might be generalised to other settings. The paper could benefit from expanding the discussion to include how differences in infrastructure, culture, or community norms might affect the scalability of the study’s approach. Ethical considerations, such as phone retention, may also be under-discussed in terms of their potential impact on the findings or on the design of the study.

Response: To address the generalizability of this research we added a section in lines 702-714 stating: “Our findings show that mobile applications using hardware on budget smartphones can generate useful data with limited interventions and suggests that mobile application development should be more widely considered in citizen science data collection efforts. Future research on the role of incentivization for data collection that utilizes technology should consider how to further disambiguate the inherent benefits in participation related to technological access, such as social status or monetary benefit from retaining equipment. We suggest that with increased mobile phone ownership, this complexity may be overcome by utilizing current phone owners, rather than providing participants with hardware.”

The focus group discussions did not include specific questions that would allow us to analyse differences in culture and community norms beyond the motivations participants stated before the trial or their experiences afterwards. Differences in attitude and community norms were addressed in the discussion lines 575-678, and we have added in lines 709-712 a call for more crossectional work that may generate the data necessary to analyse the effects of infrastructure and community norms.

Comment 6: Future Research Directions: the conclusion could benefit from a brief discussion of future research priorities, such as testing alternative incentive models, evaluating long-term engagement strategies, and/or assessing the usability of the acoustic mosquito sensors in the study context.

Response: Future research directions we have added a section to the discussion in lines 702-713: “Future research on the role of incentivization for data collection that utilizes technology should consider how to further disambiguate the inherent benefits in participation related to technological access, such as social status or monetary benefit from retaining equipment. We suggest that with increased mobile phone ownership, this complexity may be overcome by utilizing current phone owners, rather than providing participants with hardware. Additionally, as mobile phone use becomes more popular, opportunities for large scale studies will support more complex statistical approaches to account for the complex structures that underpin intrinsic motivators and responses to extrinsic motivators, such as proximity to urban areas which are potentially obscured by the inferential statistical methods used in this study.”

Comment 7: Methodology improvement: the supplementary file provides detailed documentation of participant demographics and focus group questions, which supports the reproducibility of the study. Nevertheless, the modified random walk technique used for participant selection is not sufficiently explained in either the manuscript or the supplementary materials. While the inclusion of demographic comparisons and FGD protocols is quite valuable, the lack of detail about the recruitment method limits the transparency of the methodology. For instance, how were starting points for recruitment selected? Were any randomisation procedures applied to maintain the sampling unbiased? Additionally, minor gaps in the description of the design of text message reminders could also limit reproducibility. It is recommended that the authors provide additional detail or supplementary materials for these two aspects.

Response: Details for how random walks were carried out is included in the methodology in lines 247-249, with details on reducing bias added in line 248: “To reduce the bias of entry point selection, each household was approached sequentially along the street and asked if they wanted to participate, until the number of households required to participate at the site (n=37) was met.”

Regarding the methodological details of the SMS reminders, those reminders were not a component of this study but were part of the first study in Tanzania.

Comment 8: Data availability seems to be in compliance with PLOS guidelines. Still, it is important to confirm that the repository cited by the authors includes raw data points (such as the individual participation records and anonymised FGD transcripts), as I was not able to access it. The statistical methods used appear to be appropriate, but I would recommend consulting someone with higher expertise in this matter. The methodology is sufficiently detailed overall. However, minor gaps in the description of participant recruitment (e.g., modified random walk technique) and the design of text message reminders could be further improved.

Response: The additional raw data is available now at 10.6084/m9.figshare.27332124.

Comments from Reviewer 2

This study has addressed an important question about the influence of monetary incentives on participation (data collection) in citizen research. The methodology and results have been excellently presented and discussed. The researchers have done an excellent job of using quantitative and qualitative methods to address their study questions. The manuscript is indeed of very high quality.

Here are a couple of comments that I recommend that the authors consider:

Comment 1: The topic of the study is not congruent with the study objective or question. The title “What incentives encourage local communities to collect and upload mosquito sound data by using smartphones? A case study in the Democratic Republic of the Congo” suggests that the study has evaluated different type of incentive while it has specifically assessed the influence of “monetary incentives” A title that would fit better with the study objective/question/scope could be “Do monetary incentives encourage local communities to collect and upload mosquito sound data by using smartphones? A case study in the Democratic Republic of the Congo”

Response: The title of the manuscript has been adjusted to account for this suggestion.

Comment 2: The researchers may have made efforts to adjust for and discuss confounding inherent on the research design but need to describe this more clearly in the report. Also, I recommend that authors highlight known limitations of the main inferential statistical approach used in comparative analysis.

Response: A sentence has been added in the methods to address the use of post-hoc test to reduce risk of type 1 error in line 341.

To address the limitation of inferential statistics, a section has been added to the discussion in lines 709-712, suggesting future studies use cross-sectional data that may allow for more complex modelling approaches in analysis.

---

## [Decision Letter · Decision Letter 1]

30 May 2025

PONE-D-24-49851R1Do monetary incentives encourage local communities to collect and upload mosquito sound data using smartphones? A case study in the Democratic Republic of the CongoPLOS ONE?

Dear Dr. Storer,

Thank you for submitting your manuscript to PLOS ONE. After careful consideration, we feel that it has merit but does not fully meet PLOS ONE’s publication criteria as it currently stands. Therefore, we invite you to submit a revised version of the manuscript that addresses the points raised during the review process.

We look forward to receiving your revised manuscript.

Kind regards,

Raquel Inocencio da Luz, Phd

Academic Editor

PLOS ONE

Journal Requirements:

Reviewers' comments:

Reviewer's Responses to Questions

**Comments to the Author**

Reviewer #3: All comments have been addressed

2. Is the manuscript technically sound, and do the data support the conclusions?

Reviewer #3: Yes

3. Has the statistical analysis been performed appropriately and rigorously?

Reviewer #3: Yes

4. Have the authors made all data underlying the findings in their manuscript fully available?

Reviewer #3: Yes

5. Is the manuscript presented in an intelligible fashion and written in standard English?

Reviewer #3: Yes

Reviewer #3: See attached manuscript. My main concern is that the paper found "no effect of incentives" on outcomes. However, it is not clear from paper how incentivization was implemented and measured. yes, this is "key" in the paper. Therefore, the "no effect" reported in the paper might result from a flawed design and implementation of the "incentivization process". The authors should elaborate a bit more on that. Another thing, the paper is silent on how sample sizes in the treatment and control groups were calculated. This might support the "no effect" or explain the "no effect" observed in the paper.

**Do you want your identity to be public for this peer review?** For information about this choice, including consent withdrawal, please see our Privacy Policy

Reviewer #3: **Yes: ** Professor Zacharie Tsala Dimbuene

---

## [Author Response · Author response to Decision Letter 2]

13 Jun 2025

Journal Requirements:

Comment 1: Please review your reference list to ensure that it is complete and correct. If you have cited papers that have been retracted, please include the rationale for doing so in the manuscript text or remove these references and replace them with relevant current references. Any changes to the reference list should be mentioned in the rebuttal letter that accompanies your revised manuscript. If you need to cite a retracted article, indicate the article’s retracted status in the References list and also include a citation and full reference for the retraction notice.

Response: The citation list has been reviewed and there were no instances of retracted papers. The URL links to some references were incomplete and these have now been amended. The reference list was amended with references to more completely explain the health administrative regions of the DRC and the geographic and population data of the country, with the previous references re-numbered to reflect this change.

43. United States Department of Agriculture, Foreign Agricultural Service. Congo: Democratic Republic of Congo - Country Overview. Nairobi: Global Agricultural Information Network; 2025 Feb 26. Report No.: KE2025-0004.

46. GRID3. GRID3 COD Health Areas v5.0 [Internet]. New York: Geo-Referenced Infrastructure and Demographic Data for Development (GRID3); 2023 [cited 2025 Jun 9]. Available from: https://data.grid3.org/datasets/GRID3::grid3-cod-health-areas-v5-0/explore

47. World Population Review. Kinshasa Population 2024 [Internet]. 2024 [cited 2025 Jun 6]. Available from: https://worldpopulationreview.com/cities/dr-congo/kinshasa

48. Institut National de la Statistique (INS), République Démocratique du Congo. Annuaire statistique 2020 [Internet]. Kinshasa: INS; 2021 [cited 2025 Jun 9]. Available from: UNDP‑CD mirror (PDF).

Comments from Reviewer 3

Comment 1: My main concern is that the paper found "no effect of incentives" on outcomes. However, it is not clear from paper how incentivization was implemented and measured. yes, this is "key" in the paper. Therefore, the "no effect" reported in the paper might result from a flawed design and implementation of the "incentivization process". The authors should elaborate a bit more on that.

Response: We were somewhat puzzled by this comment because the details of how incentivization was implemented are in manuscript lines 239-242 as follows: ‘Control group participants were provided with the HumBug tool (a smartphone running the MozzWear app and the HumBug bed net), and one dollar to pay for an internet connection. The incentive group was provided with the HumBug tool and one dollar for an internet connection, and an additional ten dollars each month paid via airtime to their mobile phones.’ The incentive in this trial was the ten dollars paid each month. It remains unclear to us therefore what additional information reviewer 3 requires. We also note that the other two reviewers did not find this information on incentives lacking in the paper.

Similarly, the outcomes measured in the trial are detailed in lines 122-132: ‘i) participant data collection activity during the trial period; ii) participant effort (the number of uploads per participant made during the sampling period, indicative of following trial protocols); and iii) the persistence of participation over time (whether trial participants continued to upload data throughout the sampling period). To address these questions, we compared participant activity (weeks active) and sampling effort (number of uploads) over the sampling period to assess differences in participation associated with receiving a monetary incentive for data collection. This study builds upon our previous work in Tanzania to fill gaps in understanding how monetary incentives alone influence the consistency and quality of audio data collection.’ We therefore remain unclear what more the reviewer would wish to see. Again, the other two reviewers did not find this information lacking. However, we have now added specification in line 130 that we are testing the effect of a monetary incentive on data collection to hopefully provide greater clarity.

Comment 2: Another thing, the paper is silent on how sample sizes in the treatment and control groups were calculated. This might support the "no effect" or explain the "no effect" observed in the paper.

Response: Again, we are puzzled by this comment. Sample size calculations are clearly described in methodology line 247, stating the 37 households were recruited to participate at each site with lines 341-344 detailing sample size changes due to some participants dropping out of the study as follows: ‘Some participants decided to withdraw from the trial during its course and others failed to upload data, resulting in overall participation of 33 from the control group and 31 from the incentive group in the Kinshasa trial sites (n=64), and 33 from the control group and 35 from the incentive group in Bandundu trial sites (n=68), totaling 132 participants’. These were the sample sizes used in statistical analysis. However, to further clarify, we have now also added repetition of sample sizes to figure legends, throughout the results, and in the supplement.

As requested by reviewer 3 we have now made it clear that Kinshasa is referred to as a province in this paper rather than as a city and we have further added clarifying details on the health administrative set up of the country and used this terminology throughout the manuscript.

---

## [Editor Report · Decision Letter 2]

20 Jun 2025

Do monetary incentives encourage local communities to collect and upload mosquito sound data using smartphones? A case study in the Democratic Republic of the Congo

PONE-D-24-49851R2

Dear Authors,

We’re pleased to inform you that your manuscript has been judged scientifically suitable for publication and will be formally accepted for publication once it meets all outstanding technical requirements.

Kind regards,

Raquel Inocencio da Luz, Phd

Academic Editor

PLOS ONE

---

## [Editor Report · Acceptance letter]

PONE-D-24-49851R2

PLOS ONE

Dear Dr. Storer,

I'm pleased to inform you that your manuscript has been deemed suitable for publication in PLOS ONE. Congratulations! Your manuscript is now being handed over to our production team.

Kind regards,

on behalf of

Dr Raquel Inocencio da Luz

Academic Editor

PLOS ONE